# Does Tibetan Household Livelihood Capital Enhance Tourism Participation Sustainability? Evidence from China’s Jiaju Tibetan Village

**DOI:** 10.3390/ijerph19159183

**Published:** 2022-07-27

**Authors:** Wei Shui, Yiyi Zhang, Xinggui Wang, Yuanmeng Liu, Qianfeng Wang, Fei Duan, Chaowei Wu, Wanyu Shui

**Affiliations:** 1College of Environment and Safety Engineering, Fuzhou University, Fuzhou 350116, China; shuiwei@fzu.edu.cn (W.S.); 200620011@fzu.edu.cn (Y.L.); wangqianfeng@fzu.edu.cn (Q.W.); n190620015@fzu.edu.cn (C.W.); 2Department of Geography, McGill University, Montréal, QC H4G 2Y8, Canada; yiyi.zhang2@mail.mcgill.ca; 3School of Historical Culture and Tourism, Sichuan Minzu College, Kangding 626300, China; 4College of Urban and Environmental Sciences, Peking University, Beijing 100871, China; duanfei@pku.edu.cn; 5College of Water Resource and Hydropower, Sichuan University, Chengdu 610065, China; shuiwanyu@stu.scu.edu.cn

**Keywords:** tourism participation, sustainability, livelihood capital, livelihood strategy, Tibetan

## Abstract

Identifying effective transformations to reduce poverty and approach rural sustainability is at the core of the first sustainable development goal of the United Nations. This article offers scientific support for continued efforts in sustaining rural development and livelihood resilience. Many studies have examined drivers of livelihood transition from farming to non-farm activities, especially participation in tourism against the backdrop of rural tourism development. However, few studies have identified ways to measure the level of tourism participation or have discussed how household-level capital influences decisions regarding tourism participation made by Tibetan ethnic households. This article assesses the role of livelihood capital in the adoption of tourism activities at the household level in Jiaju Tibetan Village, an ethnic region that is experiencing struggling agricultural business and developing tourism sector. Using household survey data, this study presents an ordinal logistic regression model to identify the determinants of the household tourism participation level. The results showed that households’ tourism participation was influenced by physical capital (e.g., proximity to major roads, odds ratio = 2.83 at *p* = 0.024; fixed capitals, odds ratio = 101.19 at *p* = 0.039), human capital (e.g., availability of family labor, odds ratio = 0.25 at *p* = 0.004; availability of skilled member, odds ratio = 2.91 at *p* = 0.002), and social capital (e.g., relatives in governmental sectors, odds ratio = 5.22 at *p* = 0.044; government payments, odds ratio = 8.78 at *p* = 0.04), while the influence of financial capital was not significant. The proximity to major roads, availability of skilled members, fixed assets, and direct and indirect support from the government to households were significantly and positively associated with tourism participation level. The effects of household labor availability and annual family income remain unclear. Overall, household livelihood capital plays a critical role in the enhancement of tourism participation in Jiaju Tibetan Village. Our findings have implications for understanding the shift of on-farm occupation to off-farm activities in tourism and for the pursuit of policies contributing to poverty reduction and rural revitalization in China as well as to the Sustainable Development Goals.

## 1. Introduction

In rural China, especially in remote areas with a historic reliance on traditional agriculture, peasant workers leave their home villages for cities to better their prospects, resulting in a widening gap in urban-rural development. In China, 2 million hectares of agricultural land fall out of production each year [1]. To ensure food security and rejuvenate the country as well as achieve a moderately prosperous society by 2020, the 19th National Congress called for the implementation of a rural revitalization strategy that prioritizes the development of agriculture and rural areas. Among various approaches, tourism development has been considered an effective way to strengthen urban-rural integration [1,2]. Tibet is the only provincial-level distressed area in China, characterized by mountainous features, a fragile ecological environment and a low population. It is facing inadequate development and ever-growing economic, social and ecological needs. Livelihood development in Tibetan villages touches on the UN’s 2030 Agenda for Sustainable Development which has attracted the attention of people worldwide. It is essential to explore the extent to which households in Tibetan areas depend on alternative livelihoods and to identify the key factors that influence the level of household participation.

Rural tourism is widely recognized as an effective economic growth source in response to traditional agricultural restructuring [3,4]. Tourism development provides local farmers with off-farm employment opportunities as well as important pathways out of poverty. Recent review and empirical studies have shown the positive relationships between tourism participation and community welfare [5,6]. Community-based tourism is regarded as an environmentally friendly entrepreneurial activity that creates benefits for local populations in poor agrarian contexts [7]. From a planning perspective, the institutional dimension of tourism sustainability calls for strengthening community participation in the decision-making process [8]. Local participation in tourism benefits rural sustainability by empowering households to involve in the decision-making process for local development. This suggestion is consistent with recent research demanding bottom-up initiatives be implemented to improve people’s livelihoods [1,9].

Peter E. Murphy pioneered the community approach to tourism development and encouraged host community members employed in agriculture to volunteer in tourism development [10,11]. Considerable research has explored the theoretical basis for community-based tourism [12,13] and the role of community-based tourism in poverty reduction and sustainable livelihoods [14,15]. Community-based tourism ventures aim to ensure that community members have enough power in decision-making on tourism-related projects [16]. However, the opportunities for households to participate in tourism are unevenly distributed. Several case studies and reviews demonstrate the differences in patterns and levels of tourism participation in different destination communities [17,18,19]. Especially, in developing countries, communities rarely participate in tourism planning, development, and tourism-related processes [20]. Tourism development at the collective (community) level is largely dependent on household tourism participation. Our study focuses on tourism participation decisions at the household level which is commonly used in livelihood studies and relevant for understanding community participation at a large scale.

In Jiaju Tibetan Village, where the agricultural potential is low, Pro-Poor Tourism, termed to represent tourism development that takes into account the opportunities and needs of the poor [21], has been established as the policy agenda to increase non-farm activities and involve local people in the decision-making process. To achieve these goals, the Garze government has proposed a series of tourism enhancement initiatives, such as the promotion of a “Homestay Mode”, the appraisal of “Homestay Household Models” and policy approaches such as government investment in infrastructure and house alteration. However, in our study, nearly half of the respondents reported zero involvement in tourism. Thus, it is essential to understand the underlying mechanisms that lead to the heterogeneity in tourism participation levels among local households. The factors that influence household tourism involvement are diverse and complex [22]. Some research has explored these factors using capital-related concepts such as the Sustainable Livelihood Framework (SLA)—a concept demonstrating links between different factors affecting rural livelihoods—and found that the likelihood of household-level participation in tourism is directly subject to households’ livelihood capital [23,24,25,26]. However, the mainstream literature still lacks a universally accepted measurement of household tourism participation levels, and little is known about the households’ determinants of tourism participation, especially in the western part of China to which the Chinese government has given priority in its poverty reduction efforts. A systematic analysis is needed to improve our understanding of the pathways for the equitable and sustainable development of tourism and community livelihoods. This empirical study focuses on a well-known ethnic tourism destination, Jiaju Tibetan Village, which has shown varying levels of household tourism participation, to address the gap and demonstrate how SLA can be used as a guideline to construct an innovative quantitative evaluation index system for analyzing factors that underlie the sustainability of Tibetan Village households’ livelihoods, contributing to the sustainable livelihood theory.

## 2. Study Area

Tibetan farmers are among the poorest in China, according to official data on per capita GDP, household income, and expenditure. Rural Tibet is currently shifting from a predominately agricultural economic system to a system in which earning non-farm income has become dominant [27]. There is an upsurge of interest from governments and development organizations in Pro-Poor Tourism for poverty alleviation. The Jiaju Tibetan Village, rated as “the Most Beautiful Ancient Village in China” by the magazine Chinese National Geographic in 2005, lies in the north of Danba County, east of Garze Tibetan Autonomous Prefecture. It is a well-known 4A-level tourism resort with magnificent scenery and unique ethnic architecture. The tourism industry in Danba is expanding and the visitation has rapidly increased in recent years. The annual tourist visitation increased by 65% between 2014 and 2016, from 530,000 to 876,000 (Garze Statistical Information Website, 2019).

Jiaju Tibetan Village is a graceful, idyllic place located at an altitude of 2200 m and exposed to an ecologically sensitive environment. In the village, there are more than 100 vernacular Tibetan-style blockhouses distinctive with vibrant roofs, eaves and walls. Due to the lagging infrastructure construction and mountainous landscape, Jiaju Tibetan Village is disadvantaged in agricultural productivity. Since a few tourists ventured into Jiaju Tibetan Village in the last century and local households were paid for homestay services, theme tours featuring Tibetan culture and rural life were gradually introduced. The Tibetan households, who used to rely entirely on farming as their principal income source, have become involved in and benefitted from tourism-related activities. Major tourism activities include horse-riding, specialty food tasting, exotic performance viewing, mountain tours and traditional cultural experiences oriented mostly by local residents. In 2017, approximately 100 million yuan was invested to upgrade the capacity of the Jiaju Tibetan scenic spot to provide services.

However, as tourism develops, different forms and levels of tourism participation among Tibetan households have developed, and substantial differences exist in income across interested groups. This situation has led to the conflicts inherent in livelihood choices. According to previous studies [28], households that participate in tourism derive most of their tourism revenue by providing homestay services; income from selling tourism merchandise to customers is low. Considerable discontent with benefit sharing among local households has been observed. It is necessary to create opportunities for residents to actively participate in tourism development at the right time and stage; otherwise, it may be difficult for local people to obtain adequate benefits or sustain their current share of tourism development [29]. In Jiaju Tibetan Village, the diversification of livelihoods and tourism-related job is common among Tibetan households, but the factors that influence the opportunity to actively participate are unexplored.

## 3. Methods and Data

### 3.1. Conceptual Framework

The factors that influence the level of community tourism involvement are diverse and complex. Existing literature has discussed factors that influence household participation in tourism, including the goals and characteristics of the community and the family tourism business, as well as, the institutional context in which the household is located [30,31]. Pongponrat’s [32] theory of planned behavior explained that the amount of control individuals have over their specific behaviors depends on the skills, resources and support of others. Similarly, launching a tourism business is a planned behavior for farmers, so individual and family characteristics and resource endowment can be valid predictors of the likelihood that a household participates in tourism. The Sustainable Livelihood Framework (SLA) includes main elements that can represent the capacity or resources to overcome entry barriers to livelihood activities including tourism operations. At the center of the framework are five types of capital required for a means of living: natural capital (e.g., land, water), physical capital (e.g., energy, production equipment), social capital (e.g., access to wider institutions of society), financial capital (e.g., savings, pensions), and human capital (e.g., health status, skills), on which households draw to build their livelihoods. Households’ decisions to take part in tourism are found to be influenced by the accumulation of livelihood capital [26,33,34]. This study classified tourism participation level and identified indicators of the five types of livelihood capital that may influence the level of household tourism participation (Table 1) based on SLA and studies exploring the forms and roles of different capitals in tourism development and community sustainability [25,35,36,37].

### 3.2. Variables and Hypotheses

#### 3.2.1. The Design of Dependent Variable

The tourism participation level was used as the dependent variable. Previous studies have tended to understand the tourism participation level at a larger scale by examining the overall participation ratio in communities and participants’ characteristics and overlooked different stratifications and pathways of tourism participation among households. This study looks at the heterogeneity in tourism participation levels between households. Hotels are of central importance for guests [38]. It has also been indicated that a way to address the highly uneven spatial development of tourism is to act more incisively to support families by establishing guesthouses [39]. Danba receives a large number of overnight tourists, increasing the demand for family hotels [40]. A highly developed hospitality industry is a feature of destinations that demonstrate high levels of tourism participation. Thus, households that provide homestay services reflect a high degree of tourism participation. The surveyed households were classified into three groups based on their engagement in hospitality services. The tourism participation level was defined based on the proportions of family members with self-employment in hospitality and lodging as their income source. Households were considered to present a high level of tourism participation, coded 1, when more than half of the family members served the tourist homestay. A medium level was coded as 2, indicating that less than half of the family members served the tourist homestay. In cases where no members engaged in tourism-related jobs, households were considered to present a low level of tourism participation, which were coded 1.

#### 3.2.2. The Design of the Independent Variable and Hypotheses

Table 1 lists the indicators used to calculate the quantity of the different types of capital. In an input-output analysis of China’s water consumption, the hospitality industry is recognized as a sector that indirectly increases the use of water resources to a great extent [41]. Deyà et al. [42] suggested that water resources may provide a competitive advantage for farmers entering the tourism market who intend to offer leisure and tourism services to enter the tourism market. We hypothesize that the tourism participation level is dependent on sufficient water supply (N_1_). Given that building new houses to provide homestay services also consumes timber and wood products from forests, it is hypothesized that the landholding of forests may positively influence households’ tourism participation level (N_3_). The arable land provides resource based for cultivating and harvesting crops, potentially increasing the stocks of physical assets and the possibility of the conversion of arable land to bed-and-breakfast establishments. Thus, the landholding of arable land is hypothesized to positively influence household participation in tourism (N_2_).

Physical capital refers to the facilities and fixed assets that are relatively close to the household’s production and activities. With economic growth, proximity replaces traditional geographical factors such as landform and physical resources as the major determinant of non-farm economic activities [43,44]. Infrastructure, living conditions and locational factors are also considered to shape different forms of labor allocation in rural areas [24,45]. In tourism development, households with a longer distance to roads are at a disadvantage in transporting materials and visitors, resulting in constraints for tourism participation and economic growth. Thus, we hypothesize that the greater the proximity to main roads, the more likely households are to participate substantially in tourism (M_1_). Bed-and-breakfast accommodations have developed over the years in many tourism destinations. The provision of food and room facilities is the main feature of family hotels in Danba [40]. It is hypothesized that households that own more beds and field crops may be more likely to engage in tourism and hospitality (M_2_, M_3_). The yields of Garze’s major crops including wheat, maize, and potato are used to estimate households’ ability to harvest crops and provide food. The crop yield of wheat, maize, and potato was multiplied by 3, 2, and 1 to account for the importance of these crops, based on the reported productivity and ABC inventory classification [46]. Similar to crop yields, livestock inventory is an important indicator to measure the capacity of food supply. Livestock also serves as an income source. In some studies, livestock inventory is used as an indicator to measure financial capital [47]. This study translated the livestock inventories into physical capital, using the average prices of swine (18.39 yuan per kilogram), cattle (24.92 yuan per kilogram) and chicken (27.96 yuan per kilogram) during the same period to calculate the weighted values. We hypothesize that the higher the value of livestock is, the more likely a household is to have a higher degree of tourism participation (M_4_). Since a positive relationship was found between fixed assets and the accessibility of family-owned businesses, we considered the ownership of fixed assets a valid predictor of households’ tourism participation level. Referring to the index systems generated in previous studies [48], this article defines the level of the ownership of fixed assets as the proportion of owned fixed assets of all 14 options. It is hypothesized that the higher the percentage of fixed assets owned by each household is, the more likely the household is to actively participate in tourism (M_5_).

Human capital comprises the quantity and quality of working-age family members, namely, availability, skills and education that allow individuals or households to pursue different livelihood strategies. In general, households with fewer family labor present a lower willingness in exploring labor-intensive businesses. The size of households’ labor force may influence the likelihood of tourism participation by offering self-employment opportunities. That is, the more family members there are, the more likely a household is to operate a family business in tourism (H_1_). With the increasing diversity of both tourists and workers in the tourism industry, there is also a greater need for people in the tourism industry to possess skills to adapt to the changing demands. Skilled labor proved to be relevant to the tourism industry and related sectors [49,50]. In this regard, it is hypothesized that the availability of skilled family members could also increase the level of household participation in tourism (H_2_).

Tibetan households’ financial capital is predominately from disposable cash flows and directly influences the pattern of household tourism participation. It is found that the greatest barrier limiting tourism business is the lack of financial means, especially for constructing and reconstructing buildings used for rural tourism [51]. Households with higher annual income may be more flexible in asset allocation and are more capable of developing homestay services. In contrast, households with lower annual income may have difficulty investing in the tourism business due to the lack of sufficient financial support. As a result, they may continue agricultural production or work at the bottom line of entering the tourism sector. However, some studies argued that the effect of family wealth is limited in households’ decisions to participate in tourism [30]. Thus, it is hypothesized that annual income has a significant influence on the level of household tourism participation, although the influence mechanism is unclear (F_1_).

Social capital mainly involves social networks and their derived support. A greater social network of relatives means more access to non-farm work and a reduction in dependence on agriculture [52]. Several empirical studies [53,54,55,56,57,58,59,60,61,62,63] also found that social capital is an important driver to promote community participation and resilience in tourism. In addition, there are interactions between social capital and other categories of capital [64,65]. With regard to the development of family-operated accommodation, the success factors of the tourism business have been linked to intangible relationships between households and the external social environment, including government and associations [66,67]. With indirect employment in these groups and powerful institutional support, households are less likely to confront entry barriers to establishing a business. Thus, we hypothesize that the greater the development of social networks of relatives and friends available for assistance, the more access households will have to operate tourism businesses (S_2_). If a household has income from policy subsidies when in need of money for lodging establishments, the household will participate more substantially in tourism (S_3_). Information on social capital was gathered through such questions as “Are there any relatives who are governmental officials in the family?” and “Have you ever received subsidies from the government when building new houses?” (Yes = 1, No = 0).

### 3.3. Data Collection and Analysis

The analysis is based on a questionnaire household survey conducted in 2016 in the Jiaju Tibetan Village. Using a random sampling of the entire village by household, and one questionnaire per household, a total of 60 households were visited. Households that were reluctant to answer questions were passed over. The survey questions were designed to gather information on population and employment characteristics, as well as, the ownership of various types of livelihood capital before households participated in tourism, or household conditions 5 years prior to data collection in cases where households never participated in tourism. The survey was administered by research fellows of the group who were well acquainted with the local language. In this study, a household was defined as having at least one member, regardless of whether that person is the head of the household. The survey was conducted with working-aged household members 15 years or older who were able to answer a series of relevant questions. This study ultimately included a valid sample of 60 indigenous Tibetan households in the area. The quantitative data collected were transferred to the SPSS version 22.0 computer software for statistical analysis. During the data analysis, invalid questionnaires were excluded based on missing data.

An Ordinal Regression Logistic model is frequently used when the response variable is multivariate and ordinal. In our case, it can provide information on the effect of capital on the odds of households moving into higher participation levels. The response variable was classified into 3 categories: “1” = low level, “2” = medium level and “3” = high level. In the model, the coefficients were estimated and interpreted by taking the equations in the form $\mathrm{Ln}[\frac{p\left(y\leq j\right)}{1-p\left(y\leq j\right)}]=\alpha_{j}+\overset{n}{\underset{i=1}{\sum }}\beta_{i}x_{i}$, where *j* = 1, 2, 3, representing the *j*th level of tourism participation, with y being the household tourism participation level, and *p*/(1 − *p*) being the odds of tourism participation at a certain level. Furthermore, *x_i_* represents the explanatory variables, *α_j_* is the error term of the model, and *β_i_* is the regression coefficient, which represents the effect direction and extent of the explanatory variables. We transformed the coefficients to odds ratios (OR) that can reflect the effect of a variable on the probability of the households engaged in the tourism business as their principal livelihood. To estimate the rationality and reliability of the model, we also report test statistics for the goodness-of-fit and parallel lines measures together with parameters generated by the model.

## 4. Results

### 4.1. Household Characteristics

As shown in Table 2, 53% of the sampled household heads were male and 47% were female. Most household members surveyed had little to no education. The level of education in Jiaju Tibetan Village was low in general. Households with 4 or 5 members accounted for the largest percentage of the surveyed sample. On average, 23% of people in each surveyed household had poor health status and were unable to continue in employment. Table 2 also indicates that the income gaps were relatively large—more than half (51.7%) of the surveyed households reported 0–20% of their income from tourism participation. Most households did not depend on tourism-related activities for their main source of income. Nearly half of the households did not participate in tourism. Overall, Jiaju Tibetan Village was experiencing the initiating stage of its tourism development and showed disparity in household income from tourism participation.

### 4.2. Descriptive Results

Table 3 shows the characteristics of household livelihood capital in Jiaju Tibetan Village by tourism participation level. Water supply generally met households’ demand. There was little variation in the size of arable land and forest owned by community residents, with per capita means of 0.77 acres and 0.80 acres, respectively. The average distance with distance between the household’s residential location and major roads was high in general. Each household owned 10 beds on average. In terms of human capital, there were 3 healthy laborers and 2 skilled members on average in each household. Households in Jiaju Tibetan Village had limited social capital. The minority of households had relatives serving in governmental sectors (40%) and village committees (28.3%), and only 21.7% of households received subsidies from the government.

Overall, proximity to roads, the number of beds, livestock, fixed assets, and access to government subsidies appeared to be higher for households with higher participation levels. Households in the low-level participation group showed lower mean values for most indicators, except the water supply satisfaction, per capita forest land, and access to government subsidies. In the group with high-level participation, more than half of the households had relatives serving in institutional sectors or were sponsored by the government. The medium-level group showed higher means in annual income, labor availability and per capita arable land than the other two groups.

### 4.3. Ordinal Logistic Regression Results

The results of the ordinal logistic regression are shown in Table 4. The result of testing the goodness of fit was significant at the 1% level (*p* = 0.000), proving the model’s applicability in summarizing the discrepancy between the observed values. The analysis yielded 0.084 for the test of parallel lines, which was greater than 0.05, indicating that the proportional odds assumption was met. The Nagelkerke Pseudo R-Square was 0.644, indicating an acceptable fitness of the model.

The estimated values of 3.934 and 6.675 indicate the likelihood of households being in the low-level and medium-level participation group, when other variables held constant. Seven variables in all the capital categories, except for the financial capital, were found to influence the tourism participation level at the 10% significance level or lower. Water supply and the availability of family labor showed a significant negative effect on the participation level (*p* < 0.05). The proximity to traffic arteries showed a positive influence and increased the odds of participating in tourism by 2.83 for a one-unit increase. The effects of owned forest land and arable land were not significant. The odds of high tourism participation significantly increased by 101.19-fold with a one-unit increase in fixed assets. The number of beds showed no significant impact on tourism participation. The number of skilled members was found to positively influence the participation level (*p* < 0.05). A household that had more skilled members was 2.91 times more likely to present a higher level of tourism participation. Most variables considered as social capital presented a significant impact on tourism participation. The odds ratio for the binary variable of relatives serving in governmental sectors was 5.22, indicating that a household with access to institutional support was 5.22 times more likely to engage in tourism than a household without such access. Households with subsidies provided by the government were 8.78 times more likely to be involved in tourism.

## 5. Discussion

The water supply satisfaction had a significant negative effect on the participation level, which was unexpected given that much work has concluded that tourism development increases water consumption [68,69]. A possible explanation is that households with a higher participation level perceived the water supply as inadequate largely due to the increasing water demand. The factors of owned forest land and arable land did not show significant associations with the participation level in this analysis. This may be because the effect of traditional geographic factors, including land ownership, has declined, gradually giving way to the proximity that showed a strong spatial effect in our study. Similar views were drawn in the previous study on the effects of geography such as household location on household income [43]. In the context of China’s urban-rural integration and rural revitalization, the construction of infrastructure networks for water conservancy, roads and pipelines is key to improving households’ production and living conditions, especially in China’s mountainous areas where livelihood opportunities are limited due to challenging natural conditions and lack of social resources and mobilization. To ensure the rights of the people to participate in tourism, it is essential to increase investment in infrastructure construction. Fixed assets showed a strong positive effect on tourism participation. Households with more fixed assets were more likely to participate in tourism in the form of providing homestay services. Fischer [70] found that pastoralism in rural Tibet is much more asset-based and that the asset wealth of Tibetans trumps their relative income poverty with regard to the factors influencing employment behavior. Therefore, households in Jiaju Tibetan Village may be asset-intensive and may be able to use their fixed assets as collateral in credit contracts to gain more access to the tourism business. In 2015, Danba County accounted for about 4.6% of total fixed asset investment in hospitality and ranked the third among all 20 counties in Garze (http://tjj.gzz.gov.cn/, accessed on 1 January 2019).

Interestingly but unexpectedly, the more labor was in a household, the lower the tourism participation level of the household. One possible reason for this relationship is the outmigration of young and skilled workers and the recruitment of outsiders. Similar conclusions were drawn in previous studies [9,71] on labor transformation on tourism and rural migration. In this regard, the effect of family labor availability on the level of tourism participation remains unclear. The result for the number of skilled members showed that a higher number was significantly and positively linked to the tourism participation level. One of the barriers to participate in tourism is the lack of capital and specific skills [72]. In Jiaju Tibetan Village, people with certain skills may have more access to start tourism businesses that meet tourists’ diverse demands. For instance, household members with skills in cuisine are more likely to support hotel and restaurant businesses. People with limited education and skills may find it difficult to compete with urban firms when starting rural businesses [73]. Thus, skill training is indispensable for tourism business startups and community-based tourism development.

Despite the necessity of family income to run a tourism business, the measurement of financial capital, annual income, was not a significant variable. This finding suggests that Tibetan households’ reliance on wealth for tourism participation was not significant. Similar relationships were found in Bagi and Reeder’s [30] research where the authors argued that investment in agritourism operations was not significantly limited by a farmer’s wealth. Furthermore, Eshliki [19] noted that enterprises with a higher share of fixed assets are generally more leveraged and maintain a lower stock of cash. Thus, this relationship could be a result of productive fixed assets that Tibetan households more commonly depend on. As for social capital, two explanatory variables showed a significant positive effect on the tourism participation level. Households with developed social networks at higher levels were more likely to participate in tourism businesses, whereas whether a household had relatives in regional ethnic associations had no significant effect on the tourism participation level. With direct and indirect support from the government, households are more likely to take advantage of economic and capacity enhancement opportunities [58]. Government departments should consider improving institutions and associations at the local level to ensure people’s rights to be informed, to be heard and to participate. Social networks provide not only valuable information but also financial assistance, which helps to increase the level of tourism participation. Whether a household receives government subsidies has a positive effect on the tourism participation level, indicating the potential interactions between social, political, and financial factors. Further studies may consider subcategories of the capital, such as social capital-informed financial capital and politically led social capital. Tourism involvement is found to positively influence social capital [58]. As an economic pillar in the Tibetan area, more equitable tourism participation may in turn benefit social capital development, contributing to a more collaborative and supportive society.

Capital investments play an important role in promoting the output growth of the rural economy and reduce the negative effects of rugged terrain [74]. Livelihood capital plays an important role in household tourism participation in the Jiaju Tibetan area. Remote areas inhabited by ethnic minorities are less likely to achieve rural rejuvenation in many aspects. The Tibetan Autonomous Region is the only provincial-level administration with a high poverty rate in China. Empirical evidence shows that the household livelihood capital in Garze, which shares the limited social economic development, plays an important role in the enhancement of rural sustainability. To improve these areas with thriving businesses, the tourism industry should be given particular attention. With the advent of “mass tourism”, the progress made in tourism helps to motivate the vitality of rural development and the internal force driving villagers to participate. Studies focusing on the consequences of tourism development on community capital have become common, but the influence of various capital assets on household tourism participation is not well studied. Bearing this context in mind, this study focuses on providing scientific assistance to the development of regional tourism and to propel urban-rural integration, with the view of building a society on the prospect of sustainability. One breakthrough of this study is the construction of ordinal logistic models for household survey data from Jiaju Tibetan Village to trace the degree of tourism participation to the dependencies on livelihood capital, revealing the key factors in solving uneven and inadequate participation in tourism among Tibetan households.

This study is limited in terms of the sample size and variable selection. Due to the natural terrain features of mountainous areas, and fragmented distribution and sparse population, only one typical Tibetan village was selected and a small number of households participated in the survey. The surveyed questions were not able to fully capture the impacts of multiple factors such as access to credit that was found to positively influence participation in non-farm employment and that can be compared with the effect of income on tourism participation [75]. The finding of the unclear direction of causation in water supply also led to a question related to feedbacks that may exist between the water supply and household satisfaction as well as between tourism participation level and other capital. To reduce the likelihood of two-way causality, the survey should specify the time at which the capital and perception are to be investigated.

## 6. Conclusions and Implications

Based on the SLA framework, this study explored factors influencing household tourism participation in Jiaju Tibetan area. The ordinal logistic results revealed that the characteristic differences in livelihood capital between households might lead to different levels of tourism participation. Our results showed statistically significant impacts of proximity to traffic arteries, the number of skilled household members, fixed assets, social networks and government subsidies on the level of household tourism participation. The effects of the number of family laborers and annual family income have yet to be determined.

The findings of this study can assist governments, tourism planners and households, as well as comparable regions worldwide in removing barriers to participation in tourism and in developing quality homestay products and sustainable tourism markets. Improving resilience out of poverty, contributing to rural revitalization and to the global goal of poverty eradication. The implications are presented as follows.

First, the proximity of a household to main roads increased the tourism participation level. It should be noted that for China to remain a competitive tourism destination in remote ethnic areas, investment in infrastructure, especially of transport systems, must continue to be a priority, ensuring equal access for the locals to participate and develop.

A second concern has to do with the influence of skilled members in a household. The more skilled members in a household, the more likely the household is to present a higher level of tourism participation. Training in tourism and hospitality for community-based initiatives could encourage people to develop their skills in tourism and adapt to changes and innovations in markets and tourist preferences. Government departments should consider strengthening skill training and vocational education in remote ethnic areas. This could equip rural communities to take opportunities to improve their quality of life and to avoid sinking back into poverty.

Finally, we suggest that the government focuses more on the social governance at the community level. In poor counties, new forms of social governance should be explored to improve social mobilization. Employment in rural committees and rural service personnel must meet certain standards to fully advance China’s governance, taking the lead to combat poverty and enhance livelihoods with local inhabitants.

In the process of poverty alleviation and rural vitalization, financial and non-financial capital accrues more quickly to some households than others. This may result in the augmentation of a welfare gap between households. Thus, the issues above should be addressed to achieve rural equity and sustainability. In pursuing China’s rural revitalization, addressing issues related to the poor ethnic areas are fundamental, and the livelihood enhancement of Tibetan households helps to strengthen the resilience of rural development as they play a decisive role in building a moderately prosperous society and achieving the United Nations’ sustainable development goals.

## Figures and Tables

**Table 1 ijerph-19-09183-t001:** Names and definitions of variables.

Types of Variables	Variable	Definition
Dependent variable	Tourism participation level	1 = low level, 2 = medium level, 3 = high level
Natural capital	Water supply satisfaction degree (N_1_)	1 = not satisfy, 2 = basically satisfy, 3 = very satisfy, 4 = completely satisfy
	Arable land (N_2_)	Per capita arable land
	Forest land (N_3_)	Per capita forest land
Physical capital	Proximity to main roads (M_1_) Accommodation capacity (M_2_)	1 = far away from main roads 2 = within 50 meters away from the main roads 3 = next to the main roads The number of beds
	Field crops (M_3_)	kg, wheat (M_31_), maize (M_32_), potato (M_33_) M_3_ = 3 × M_31_ + 2 × M_32_ + 1 × M_33_
	Livestock (M_4_)	Swine (M_41_), cattle (M_42_), chicken and duck (M_43_) M_4_ = 459.75 × M_41_ + 2492 × M_42_ + 69.9 × M_43_
	Fixed assets (M_5_)	The proportion of owned fixed assets of all 14 options
Human capital	The availability of family labor (H_1_)	Members over 15 years of age, with qualified health status (equal to or better than average)
	The availability of skilled member (H_2_)	Family members with at least one specific skill
Financial capital	Annual income (F_1_)	Annual family income in total
Social capital	Relatives in governmental sectors (S_3_)	0 = No, 1 = Yes
	Relatives in village committee (S_2_)	0 = No, 1 = Yes
	Government payments (S_3_)	0 = No, 1 = Yes

**Table 2 ijerph-19-09183-t002:** Summary of household socio-demographic and tourism participation characteristics.

Characteristics	Percentage	Characteristics	Percentage
Socio-demographic Characteristics		Annual income	
Gender		0–20,000	18.6
Male	53.3	20,001–40,000	35.6
Female	46.7	40,001–60,000	18.6
Education Level		60,001–80,000	6.8
Uneducated	30	80,001–100,000	10.2
Primary	36.7	>100,000	10.2
Secondary	23.3	Tourism participation characteristics	
High school or associate degree	6.7	The number of tourism participants	
Higher education	3.3	0 person	43.3
Household size		1 person	11.7
0–1 person	0	2 people	33.3
2–3 people	13.6	3 people	8.3
4–5 people	47.5	4 people	3.3
6–7 people	37.3	Income from tourism participation	
8 or more people	1.7	0–20%	51.7
Health status		20–40%	15
Excellent	68.3	40–60%	15
Good	1.7	60–80%	10
Average	6.7	80–100%	8.3
Poor	23.3		

**Table 3 ijerph-19-09183-t003:** Descriptive statistics for all categories of variables analyses.

Household Type	Means or %			
	Entire Household Samples (*n* = 60)	High Level (*n* = 15)	Medium Level (*n* = 17)	Low Level (*n* = 24)
Water supply	2.02	1.53	1.82	2.45
Per capita arable land	0.77	0.76	0.82	2.45
Per capita forest land	0.77	0.76	0.82	0.74
The number of beds	10.17	11.46	11	9.54
Proximity to main roads	2.23	2.6	2.47	1.88
Field crops	4361.25	3291	4532.35	4270.83
Livestock	2999.13	3405.85	3352.42	2389.92
Fixed assets	0.37	0.46	0.34	0.33
Family labor availability	3.08	2.87	3.59	2.83
The number of skilled member	1.98	2.4	2.41	1.42
Annual income	57,838.98	54,800	89,388.24	35,870.83
Any relatives in governmental sectors (%)				
Yes	40	80	35.29	65.38
No	60	20	64.17	34.62
Any relatives in village committee (%)				
Yes	28.3	60	58.82	84.62
No	71.7	40	41.18	15.38
Any government payments (%)				
Yes	21.7	86.67	88.24	65.38
No	78.3	13.33	11.76	34.62
Tourism participation level	1.81	3	2	1

**Table 4 ijerph-19-09183-t004:** Results of the ordinal logistic regression model on tourism participation level.

Variable	Coefficient	*p*-Value	Odds Ratio
Tourism participation level			
[Low level = 1]	3.934	0.090	
[Medium level = 2]	6.675	0.007	
[High level = 3]	0 ^a^		
Water supply satisfaction degree	−1.540	0.002 ***	0.21
Arable land	0.702	0.524	2.02
Forest land	1.735	0.133	5.67
Proximity to main roads	1.041	0.024 **	2.83
The number of beds	0.054	0.264	1.06
Field crops	0.052	0.592	1.00
Livestock	0.000	0.091 *	1.00
Fixed capitals	4.617	0.039 **	101.19
The availability of family labor	−1.401	0.004 ***	0.25
The availability of skilled member	1.068	0.002 ***	2.91
Annual income	0.019	0.290	1.02
Any relatives in governmental sectors			
[No = 0]	1.652	0.044 **	5.22
[Yes = 1]	0 ^a^		
Any relatives in village committee			
[No = 0]	−0.620	0.454	0.54
[Yes = 1]	0 ^a^		
Any government payments			
[No = 0]	2.172	0.040 **	8.78
[Yes = 1]	0 ^a^		
Goodness-of-Fit			
−2 Log Likelihood	73.551		
Chi-Square	47.171		
*p*-value	0.000		
Test of parallel lines			
−2 Log Likelihood	51.811		
Chi-Square	21.741		
*p*-value	0.084		

***, **, and * show that the coefficient is significantly different from zero at the 1 percent, 5 percent, and 10 percent level, respectively. ^a^ Parameters of reference groups were set as 0.

## Data Availability

Not applicable.

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
