# Peer review of "Does Tibetan Household Livelihood Capital Enhance Tourism Participation Sustainability? Evidence from China’s Jiaju Tibetan Village"

_ijerph, 2022, doi:10.3390/ijerph19159183_

Round 1

Reviewer 1 Report

Congratulation!

Good design, correctly made, good statistic and conclusions in according with the results.

The abstract is too descriptive. You have to put in the abstracts some concrete results such us ...OR values whit or without p values.

For example: The results showed statistically significant impacts of proximity to traffic arteries (OR=2.83 at p = 0.024), the number of skilled household members (OR=2.91 at p = 0.002), fixed assets...., social networks and government subsidies on the level of household tourism participation ...You have to put in the abstract in order to became more attractive.

Reviewer 2 Report

Nicely written paper overall.

Abstract is solid

Introduction does a nice job of setting up the paper. One could say there could be additional citations possible to works on sustainable tourism, community-based ecotourism, and/or social capital and tourism, but otherwise the theoretical content is soundly developed even within the limited citations provided here.

I have worked on these topcis for years, and I have never heard of Peter E. Murphy. The authors do not cite any writings of Murphy. Please reference the related materials.

The statement that “the shortcomings of studies on tourism participation is that the unit of analysis is…an individual pixel” does not make sense. Pixels refer to image resolution, and most research on resident participation in tourism is not based on imagery. It would be better to not mix metaphors like this. Furthermore, claims like this need substantiation. Again, I’ve done work in this area (not cited here) that does not rely on “pixels” or “aggregation of community units.” What the authors carry out at the household level could very well be the most common unit of analysis in studies of tourism participation. If the authors disagree, they need to provide stronger substantiation of such claims.

Pro-Poor Tourism (written as a proper noun) needs definition and citations.

Garze government – why can only a select few households be involved? Need better explanation there.

Study Area and Data Sources

It is fine to have a separate section on study area, but data sources is nearly always part of the Methodology. I suggest moving data sources into the section that follows on Methodology.

In terms of study area, the forms of tourism taking place there are not well described. What do tourists do there? What kind of tourism is taking place? Is it simply rural community tourism, such as that described here:

Xue, L., Kerstetter, D., & Hunt, C. (2017). Tourism development and changing rural identity in China. Annals of Tourism Research, 66, 170-182.

Please clarify the forms of tourism (i.e., what tourists actually do) that occur.

The concepts and methodology

Again, there is some structural/organizational confusion here. The conceptual/theoretical information should be presented in a proper literature review section following the Introduction (and followed by the study area description). Afterward, a Methods section should be presented. There you can include the study site description as well as the details about data collection instruments and analysis plans.

Unless you provide documentation, then YES, there is doubt about who is poorest and where the greatest poverty exists in China. You must cite information to substantiate any such claims about Tibetan farmer.

Survey Data

What is a whole-scale questionnaire? Explain.

The survey was gathered in 2016. That is six years ago. Is it still relevant? Hasn’t a lot occurred in the past six years that would dramatically influence the hypothesized relationships here?

Even if each household was a target sample, you must indicate what proportion of households actually completed the survey. Later it says “all Indigenous households…were visited. Are all households in the village Indigenous, or were only the Indigenous households visited? What proportion of overall household numbers is this?

Who are the research fellows of the group?  Which group?

It would be helpful to know more about the survey contents, especially how information related to the “capitals” were gathered.

How were nonsensical responses identified?

Conceptual framework

The authors start this section by introducing agritourism, something not addressed previously. They thus do not stay focused on one particular type of tourism. Rural community, communit-based tourism, agritourism, and other forms of tourism are all discussed. It would be better to first discuss how scholars have treated and defined each of these forms of tourism before settling on one label for the form of tourism most present in the study site. The focus should remain on that form of tourism for the remainder of the paper.

Given the focus on the capitals framework, there are many additional papers both in and outside of tourism studies that might be of interest to the authors that could be cited here – many of which are conspicuously absent here: 

Aldrich, D.P. and Meyer, M.A. (2015) ‘Social capital and community resilience’, American

Behavioral Scientist, Vol. 59, No. 2, pp.254–269.

Bourdieu, P. (1986) ‘The forms of capital’, Handbook of Theory and Research for the Sociology of Education, pp.241–258.

Coleman, J.S. (1988) ‘Social capital in the creation of human capital’, American Journal of

Sociology, Vol. 94, S95–S120.

Diedrich, A., Benham, C., Pandihau, L. and Sheaves, M. (2019) ‘Social capital plays a central role

in transitions to sportfishing tourism in small-scale fishing communities in papua new

Guinea’, Ambio, Vol. 48, No. 4, pp.385–396.

Gittell, R. and Vidal, A. (1998) Community Organizing: Building Social Capital as a Development

Strategy, Sage Publications, London.

Grootaert, C. (1998) Social Capital: The Missing Link?, Social Capital Initiative Working Paper 3.

Guo, Y., Zhang, J., Zhang, Y. and Zheng, C. (2018) ‘Examining the relationship between social

capital and community residents’ perceived resilience in tourism destinations’, Journal of

Sustainable Tourism, Vol. 26, No. 6, pp.973–986.

Hunt, C.A., Durham, W.H. and Menke, C.M. (2015) ‘Social capital in development: bonds,

bridges, and links in osa and golfito, Costa Rica’, Human Organization, Vol. 74, No. 3,

pp.217–229.

Jones, S. (2005) ‘Community-based ecotourism: the significance of social capital’, Annals of

Tourism Research, Vol. 32, No. 2, pp.303–324.

Levien, M. (2015) ‘Social capital as obstacle to development: brokering land norms, and trust in

rural India’, World Development, Vol. 74, pp.77–92.

Marcinek, A.A. and Hunt, C.A. (2015) ‘Social capital, ecotourism, and empowerment in shiripuno, Ecuador’, International Journal of Tourism Anthropology, Vol. 4, No. 4, p.327.

Moscardo, G., Konovalov, E., Murphy, L., McGehee, N.G. and Shurmann, A. (2017) ‘Linking

tourism to social capital in destination communities’, Journal of Destination Marketing and

Management, Vol. 6, No. 4, pp.286–295.

Onyx, J., Edwards, M. and Bullen, P. (2007) ‘The intersection of social capital and power: an

application to rural communities’, Rural Society, Vol. 17, No. 3, pp.215–230.

Park, D.B., Lee, K.W., Choi, H.S. and Yoon, Y. (2012) ‘Factors influencing social capital in rural

tourism communities in South Korea’, Tourism Management, Vol. 33, No. 6, pp.1511–1520.

Szreter, S. and Woolcock, M. (2004) ‘Health by association? Social capital, social theory, and the

political economy of public health’, International Journal of Epidemiology, Vol. 33, No. 4,

pp.650–667.

Woolcock, M. and Narayan, D. (2000) ‘Social capital: implications for development theory,

research, and policy’, The World Bank Research Observer, Vol. 15, No. 2, pp.225–249.

The definition of the variables is a bit dubious. For instance, as presented here, it is not clear how “distance from traffic arteries” serves as an indicator of natural capital.

Financial capital has but a single indicator. What about access to credit/loans?

The table suggests “government payments” is a form of social capital. That does not seem appropriate. It seems more like an indicator of financial capital. Please clarify.

The following statement is both awkward and incorrect – “In the input-output analysis of China’s water 232 consumption, the hospitality industry is recognized as a sector that fuels water resources 233 to a great extent.” Fueling water resources implies that tourism adds to water resources. Either a word is missing or this needs to be restated.

The hypotheses are a bit arbitrary and not connected to the earlier conceptual review. Why were these chose? Be sure to provide theoretical rationalization of your choices.

H4: More distance from roads = better participation in tourism? This needs better explanation.

H5: households that own more beds will be more likely to engage in tourism. This hypothesis has very strong potential for confusing correlation with causation. Isn’t having more beds a direct function of being involved in tourism? Not sure this can be tested with the present data. You can demonstrate correlations but not causation here. The same may be true with many of the hypotheses presented.

This material is not properly explained – few readers unfamiliar with the study region will be capable of interpreting this section:

“An analysis of Garze’s rural economic conditions proposed the stabilization of the cultivated area of wheat, maize and potato in the first season of 2017.” Thus, fundamental living grains, including wheat, maize and potato, are considered the measurements of the variable of field crops. As the crop yields of these 3 types can be observed as potato > maize > wheat, this article determined the weights of wheat, maize and potato as 3, 2, and 1, respectively, through the ABC inventory classification [43], which could compensate the bias towards each class.”

Additionally, all hypotheses appear to be one-way (e.g., having these livelihood assets will lead ot greater involvement in tourism). What about the other direction – doesn’t greater involvement in tourism also create or strengthen existing stocks of capital? This two-way relationship does not seem clearly articulated or maintained throughout the paper.

RESULTS

Why does Table 3 not include an ANOVA of the three groups?

Why is ordinal logistic regression needed here instead of several simple linear regressions or an ANCOVA? This does not seem to be an appropriate statistical analysis plan here.

Table 4 – p values should be reported rather than 1, 5, and 10% values.

DISCUSSION

The discussion is a bit hard to consider as the statistics carried out do not seem appropriate for testing the hypotheses laid out earlier.

In any case, the additional literature provided above will provide additional contexts for the statistical results found here. I would encourage incorporating at least some of that content into the Discussion.

Statements such as the following are borderline racist:  “Empirical evidence shows that, the household livelihood capital in Garze, which shares the same backward degree of social economic development, is likely to contribute to the enhancement of rural sustainability.” I added emphasis to the offending portion. This is victim blaming and more a function of structural societal disadvantage and oppression rather than any particular degree of “backwardness” of the rural population. The authors are strongly encouraged to reconsider the use of any such pejorative language or tone used to characterize rural regions or their populations!

Again, as noted earlier, remain cognizant of the two-way relationship between capitals and tourism. The Discussion section would be a good place to address this issue and the limitations of the present study.

CONCLUSION

The first line of the conclusion is horrible because the “DFID” and “SLA” are acronyms that have not been defined in the paper. They must be defined prior to appearing in the text.

Reviewer 3 Report

Does Tibetan Household Livelihood Capital Enhance Tourism Participation Sustainability? Evidence From China's Jiaju Tibetan Village

Comments:

· The Abstract is poorly written and needs to be revised and summarized according to the study's purpose, methodology, and main findings.

· It would be better to read and look at the paper (Liu et al., 2022).

· Author should explain its theoretical contribution.

· Methodology section needs to be revised.  

· The author should check the whole paper's grammatical errors and English proofreading.

This paper needs more improvement in the context of its theoretical contribution and particle implication. Also, this paper needs to improve its methodology section. Last but not least, in the conclusion part Author mentioned that “Based on the DFID’s SLA framework”, but I didn’t see anything related to this framework in this article. So authors should revise the whole manuscript and read the currently published paper by Liu et al, 2022 for further reference.

Liu, Y., Shi, H., Su, Z., & Kumail, T. (2022). Sustainability and Risks of Rural Household Livelihoods in Ethnic Tourist Villages: Evidence from China. Sustainability14(9), 5409.

Round 2

Reviewer 2 Report

I would still like to say that the authors are confusing conceptualization with operationalization. Conceptualization and the associated theoretically-driven hypotheses should all be derived from the existing literature and thus located within the Literature Review section. Operationalization, how variables are specifically defined for a given study, can then be located within the Methods section. This issue need not hold up publication here as the sections are easy to locate, but in the future, the authors may avoid similar feedback and delays in publication if they keep conceptual development in the literature review and limit the content in their methods sections to specific details of operationalization of variables.

An example might clarify this concern. I did not find the explanation of why government subsidies qualify as social capital compelling. Subsidies are not part of the “value of social relations,” as the social capital concept is typically defined. Thus, conclusions drawn on the basis of that operationalization of social capital are likely to be flawed and disconnected from prior conceptualization of social capital in the literature.

Nevertheless, the authors have improved the manuscript considerably based on the other feedback provided. I am prepared to support the paper’s publication.